# The Mating Type (MAT) and Virulence of *Sporothrix schenckii sensu stricto* Isolates Maintained in Culture Collection

**DOI:** 10.3390/microorganisms11092335

**Published:** 2023-09-18

**Authors:** Thais Barreira, Danielly Corrêa-Moreira, Cintia de Moraes Borba, Rodrigo Caldas Menezes, Aurea Maria Lage de Moraes, Manoel Marques Evangelista Oliveira

**Affiliations:** 1Laboratory and Facility Multi-User, Evandro Chagas National Institute of Infectious Diseases, FIOCRUZ, Rio de Janeiro 21040-360, Brazil; thais.guimaraes@ini.fiocruz.br; 2Laboratory of Taxonomy, Biochemistry and Bioprospecting of Fungi, Oswaldo Cruz Institute, FIOCRUZ, Rio de Janeiro 21040-360, Brazil; cintiaborba@terra.com.br (C.d.M.B.); aurea@ioc.fiocruz.br (A.M.L.d.M.); 3Laboratory of Clinical Research in Dermatozoonoses in Domestic Animals, Evandro Chagas National Institute of Infectious Diseases, FIOCRUZ, Rio de Janeiro 21040-360, Brazil; rodrigo.menezes@ini.fiocruz.br

**Keywords:** *Sporothrix schenckii sensu stricto*, preservation in culture collection, mating type, virulence/pathogenicity

## Abstract

In an attempt to determine the mating type of different *Sporothrix schenckii sensu stricto* isolates that remained viable after a long period of preservation in a culture collection and to correlate them with the degree of virulence/pathogenicity, a PCR technique using primers designed for the sequences of *MAT1-1-1* and *MAT1-2-1* genes and a murine experimental model were used. The results showed that there was no correlation between the mating type and virulence among the isolates. Furthermore, different degrees of virulence/pathogenicity, ranging from high to low, were found among them based on different virulence parameters. It was assumed that the long period of preservation favored the changes, yielding the isolation of variants. Thus, we believe that new technologies for studies on factors can improve our knowledge of the pathogenesis of sporotrichosis.

## 1. Introduction

*Sporothrix schenckii sensu stricto* belongs to the division Ascomycota, class Pyrenomycetes, order Ophiostomatales, family Ophiostomataceae [1,2] and is one of the pathogenic species that causes sporotrichosis, a subacute or chronic fungal infection of the cutaneous and subcutaneous tissues widely distributed throughout the world, which reaches hyperendemic proportions in some regions of Brazil and the world [3,4,5,6].

Sexual identity in fungi is controlled by the *MAT* loci, which regulates the expression of pheromone and pheromone-receptor genes [7]. It is known that *Sporothrix* sp. is a heterothallic ascomycete that has two-form mating-type loci or idiomorphs called *MAT1-1* and *MAT1-2* [8]. The first study to determine the mating type genes of *Sporothrix schenkii sensu stricto* was performed by Kano et al. [9] and in the following year, the same authors confirmed the difference between the *MAT1-1* and *MAT1-2* loci, suggesting that the fungus is heterothallic [10]. 

Some authors affirm that the process of sexual reproduction in fungi has been linked to the virulence and the mating type has been found to be a source of variation in virulence [11], and it has been demonstrated that there is a significant difference in pathogenicity between *MAT1-1* and *MAT1-2*, with *MAT1-1* having greater pathogenicity on average [12]. However, to the best of our knowledge, only one published study stated that the mating type is very unlikely to be associated with the virulence of the *Sporothrix* species [13].

Microbiological collections are considered important tools for obtaining information and have evolved over time due to the growing demands of different scientific areas [14]. Currently, their function is the acquisition, preservation, identification, cataloging and distribution of authenticated microorganisms to support scientific research, epidemiological studies and the development of bioproducts among others [15]. 

In a previous study, our group evaluated the viability of 34 isolates identified as *S. schenckii* maintained for long periods of time under mineral oil at the Culture Collection of Filamentous Fungi of Oswaldo Cruz Institute/Fiocruz (CCFF-IOC). Seven of them were recovered and examined for stability of the morphological and physiological pattern [16]. Therefore, the aim of this study was to increase knowledge about these isolates regarding the identification of mating-type genes as well as the capacity for infection in an experimental model and to attempt to correlate them.

## 2. Materials and Methods

### 2.1. Isolates

Seven isolates of *S. schenckii sensu stricto* preserved in the Culture Collection of Filamentous Fungi of Oswaldo Cruz Institute, Fiocruz (CCFF-IOC) and recently reidentified by polyphasic identification [16] were used in this study (Table 1). All the isolates were originally isolated in Rio de Janeiro/Brazil, prior to the zoonotic epidemic in this city, from lesions of patients with sporotrichosis, except for IOC 1799 clinical isolate, which was a sample of Japanese origin. They were kept on potato dextrose agar—PDA (Becton, Dickinson and Company, Sparks, MD, USA), at 30 °C for later experiments.

### 2.2. PCR Conditions

The disruption of the fungal cells to the extraction of genomic DNA of *S. schenckii sensu stricto* isolates was performed by grinding in liquid nitrogen, with TES (Tris, pH 7.5—10 mM, Ethylenediaminetetraacetic acid (EDTA), pH 8.0—10 mM, Sodium dodecyl sulfate—SDS 0.5%) buffer (in house). To purify the DNA, chloroform was used with the isoamyl alcohol methodology [11,17]. The identification of types of MAT idiomorphs was performed according to the protocol previously described by [1], with some adjustments, using a PCR with primers designed for the idiomorphic sequence of the gene *MAT1-1-1* (MAT1_1_1F, AAGCACGCCAGTTTCATTCT; MAT1_1_1R, CACCAACGAGCATCTCATGT), and gene *MAT-1-2-1* (MAT1_2F, GATCTCAACGGCCATCTTGT; MAT1_2R, GCTACATACTTCCGCCCTGA). We added 75 ng of DNA template and a 10 µM concentration of each primer in a total volume of 50 µL. The amplification program included 35 cycles and an annealing temperature of 55 °C. Amplifications were performed in a Veriti TM 96-Well Thermal Cycler (Thermo Fisher Scientific, Waltham, MA, USA). PCRs were performed separately for each idiomorph. Amplification was confirmed by visualization of the amplicons in agarose gel electrophoresis 2%, DNA ladder molecular marker, 100 bp (Promega Corporation, Fitchburg, WI, USA) at a current of 70 volts for 1 h.

### 2.3. Experimental Sporotrichosis

#### 2.3.1. Inoculum

*S. schenckii sensu stricto* isolates were inoculated in liquid Sabouraud medium at 25 °C under constant agitation (120 oscillations per min). After 11 days, the fungal growth was filtered through sterile gauze and conidia were pelleted by centrifugation at 3000× *g* for 5 min. After 3 washes, they were resuspended in 1 mL of PBS (50 mM phosphate-buffered saline at pH 7.2) and counted with a Neubauer chamber. Cellular viability was determined by the colony forming unit (CFU) [18], resulting in more than 85% viability. 

#### 2.3.2. Reactivation of the Isolates

The conidial suspension obtained as described in Section 2.3.1 was inoculated subcutaneously at the base of the tail of three BALB/c mice (from Science and Technology in the Biomodels Institute, ICTB/Fiocruz, Rio de Janeiro, Brazil) for each isolate, at a concentration of 3 × 10^6^ conidia in 0.02 mL of sterile PBS per isolate. After 30 days, or after the appearance of lesions, mice were submitted to euthanasia by an anesthetics overdose (ketamine—90 mg/kg and xylazine—10 mg/kg), and subsequently necropsied to recover the isolates from spleen culture in Mycosel Agar (Becton, Dickinson and Company, Sparks, MD, USA) at 30 °C.

#### 2.3.3. Animal Experimentation

The isolates recovered from the spleens of mice were inoculated in 128 male BALB/c mice (16 for each fungal isolate), aged 6–8 weeks, weighing approximately 25 g, following the same infection protocol described in Section 2.3.2. 

The animals were divided into eight groups containing 16 mice each: (a) seven groups infected with (one per *S. schenckii sensu stricto* isolate): IOC 1275, IOC 1799, IOC 1835, IOC 1912, IOC 2547, IOC 2835 and IOC 2993; and (b) control group inoculated with PBS. 

In agreement with the statement of the ethics committee, the minimum number of animals per species were used, to ensure the recovery of fungal cells. 

##### Infection Capacity of the Isolates

Analysis was performed following the protocols described by Corrêa-Moreira et al. [19]. At each point of observation (21, 35 and 49 days after inoculation), clinical signs were observed such as cutaneous lesions, alopecia and apathy in the mice. After euthanasia, the mice were weighed and the macroscopic examination of the internal organs was performed, so the skin, spleen, lungs, heart and kidneys were removed from all the mice. 

The spleen and body weight ratios of each infected mouse and of control mice were also determined. The splenic index was calculated based on the ratios of the relative weight of the spleens from infected mice expressed as units in relation to the control. The mean value for the relative weight of spleens in each control group of mice was equal to one unit.

To determine the number of CFU, the spleens were weighed and homogenized in a sterile, complete RPMI-1640 medium containing L-glutamine (Sigma Chemical, St Louis, MO, USA) and the suspension was adjusted to 2 mg of tissue per ml and samples of 150 µL of each homogenate were transferred to Petri dishes with Mycosel agar (Becton Dickinson and Company, Sparks, MD, USA), incubated at 37 °C for 15 days, for fungal reisolation and quantification of the colony forming unit.

The survival assessment was performed by observing mice for 50 days following inoculation. Mortality in each group was noted daily. 

To observe histological alterations, the organs collected (skin, heart, kidney, lungs) were immediately fixed in 10% buffered formalin and embedded afterwards in paraffin, sectioned and placed on slides, and stained with hematoxylin-eosin and Grocott’s methenamine silver (Both from Proc9 Company, Canoas, Brazil), to confirm the presence of the fungus in the tissues. Briefly these staining protocols are described below. 

For GMS staining, the sections were put into distilled water and oxidized with 4% aq chromic acid at room temperature for 1 h. After, they were washed and treated with 1% sodium metabisulphite for 1 min, followed by 3 washes under running tap water and rinsed thoroughly in distilled water for 3 min. After this step, the sections were placed in a pre-heated working silver solution in a water bath at 60 °C for 15 to 20 min until the sections turned yellowish-brown, then they were rinsed again, and toned with 0.2% gold chloride for 2 min. After rinsing, the sections were treated with 2% sodium thiosulphate for 2 min, washed with running tap water for 5 min, counterstained in working light green for 15 sec, the excess light green was rinsed off the slide with alcohol, then the sections were dehydrated, cleared and mounted. To HE stain, the sections were stained with alum haematoxylin for 4 min, rinsed, differentiated with 0.3% acid alcohol by two quick dips, rinsed in tap water and stained with eosin for 2 min. After, the sections were dehydrated, cleared and mounted.

#### 2.3.4. Ethical Statement

All experiments involving animal experimentation were conducted according to the approval of the Animal Use Ethics Committee of the Oswaldo Cruz Institute (CEUA-IOC permit number L-019/2020).

#### 2.3.5. Statistical Analysis

Statistical analyses were conducted in Prism GraphPad for Windows version 8 (GraphPad Software, San Diego, CA, USA). Results were expressed as (interquartile range) mean ± standard deviation (SD). For the analysis of the CFU, weight variations and splenic index, analysis of variance (ANOVA) was performed. Kaplan–Meier curves with Log-rank tests were used to describe the survival of mice. The *p*-value < 0.05 was adopted for all tests (*n* samples = 4). 

## 3. Results

### 3.1. Mating-Type Gene Analysis 

Using primers targeting *MAT 1-1* and *MAT 1-2* loci, it was possible to observe the presence of two fragments suggesting the existence of two sexual idiomorphs among isolates (Figure 1). The *MAT1-1-1* specific primers amplified a ≅ 600 bp fragment in the isolates IOC 2993 and IOC 1912 and the *MAT1-2-1* primers amplified a ≅ 450 bp fragment in the isolates IOC 1275, IOC 2835, IOC 1799, IOC 2547 and IOC 1835.

### 3.2. Experimental Sporotrichosis

#### 3.2.1. Clinical Signs

At 21 days after inoculation, macroscopic lesions were observed as little nodules at the point of inoculation of all animals inoculated with IOC 1275, IOC 2547 and IOC 1835 isolates and only one mouse inoculated with IOC 2835. At 35 days, all the animals inoculated with all the isolates presented lesions at the point of inoculation, except for those infected with the IOC 1799 isolate. Of these isolates, six of them (IOC 1275, IOC 1835, IOC 1912, IOC 2547, IOC 2835, IOC 2993) caused enlarged ulcerated lesions from the point of inoculation. Edema was observed at the site of inoculation, as well as ulcerated crusty and sporotrichoid lesions on the skin tissue in all animals inoculated with the IOC 1275 isolate, while the other isolates produced lesions in only one of the inoculated animals (Figure 2). At 49 days after inoculation, no more lesions were seen on the skin of the animals. Alopecia on body parts other than those with lesions and prostration were not observed. The animals stayed active throughout the experiment.

The examination of the internal organs showed that heart and lung macroscopically visible lesions were seen in mice inoculated with IOC 1275 at 35 days of inoculation. Mice inoculated with IOC 1835 isolates showed lesions only in the lung at the same time. On the 49th day after inoculation macroscopic lesions were not observed in different organs. 

#### 3.2.2. Weight Variation

The animals inoculated with all *S. schenckii stricto sensu* isolates had their weight increased over the evaluated time like the control group (Table 2).

No statistical difference was observed between the infected groups and control group.

#### 3.2.3. Splenic Index

Table 3 shows the splenic index of BALB/c mice after inoculation with *S. schenckii stricto sensu* isolates at 21, 35 and 49 days of infection. The mean of the splenic index values revealed the presence of splenomegaly in all animal groups. 

At 21 days after inoculation, the mean spleen weight of mice infected with the IOC 1835 isolate was higher than in other groups followed by the mean of the group infected with the IOC 2547 isolate. At 35 days after inoculation, there was no significant difference among the groups, however, as observed at the initial point, the mice infected with the IOC 2547 isolate showed greater splenomegaly compared to the other groups and maintained higher index values among the averages until the end of the experiment (49 days). No statistical difference was observed among the infected group of mice.

#### 3.2.4. Recovery of Fungal Cells

The recovery of viable fungal cells from the spleen of mice is shown in Figure 3. At 21 days of infection, no fungal cells were recovered from the spleens of the inoculated mice. On day 35 post-infection, cells were recovered from the spleens of animals infected with IOC 1275, IOC 1835, IOC 2993, IOC 2835 and IOC 1912 isolates, highlighting that the highest number of fungal cells isolated from the spleen were from mice inoculated with the IOC 2835 isolate. At 49 days after infection, it was only possible to recover fungal cells from spleens of mice infected with the IOC 1275 isolate. No statistical differences were observed between the groups. 

#### 3.2.5. Survival of Mice Inoculated with *S. schenckii stricto sensu* Isolates

The survival rates of the mice inoculated are shown in Figure 4. All animals infected with isolates IOC 1275, IOC 1799, IOC 2547 and IOC 1912 and the control group survived until the end of the experiment. On the other hand, the IOC 1835 isolate was able to cause a mortality rate of 25% of the infected mice, as well as the IOC 2993 isolate; however, the mortality induced by the first one occurred during the early time of the experiment (24 days after infection), unlike IOC isolate 2993, which caused mortality on the 42nd day of infection. The percentage of mortality induced by the IOC 2835 isolate was the highest among the groups (50%) at the end of the observation period, between the 48th and 50th days after infection. No statistical differences were observed among the groups.

#### 3.2.6. Histopathology

Table 4 shows an overview of the data on histological alterations caused in mice inoculated with different *S. schenckii stricto sensu* isolates at the predetermined times for the analyses.

At 21 days after inoculation, no histological alterations were observed in the evaluated organs. GMS staining evidenced the presence of fungal structures in the pulmonary tissue of the mice infected with IOC 1912, IOC 1835 and IOC 2547 isolates (Figure 5).

After 35 days of inoculation, significant alterations were observed in the skin of all infected groups, except for the group of animals inoculated with the IOC 1799 isolate. In the lungs of mice infected with IOC 1835, IOC 1275 and IOC 2547 isolates, significant tissue alterations were observed. In the heart of one mouse infected with the IOC 1275 isolate two foci of dystrophic mineralization in the epicardium were observed (Figure 6). 

Figure 7 shows fungal structures observed in the lung and skin GMS preparations of mice infected with IOC 1912 and IOC 1835 isolates. Although histological alterations were not detected in the tissue from mice infected with the IOC 1799 isolate, fungal structures were observed in the lung.

At the end of the evaluation period (49 days), no tissue damage in the evaluated organs and/or the presence of *S. schenckii stricto sensu* yeast cells were observed in infected mice. The control group did not present alterations and fungal cells at different times.

Our group, in an attempt to improve the accuracy in comparing the fungal isolates in terms of infection capacity and consequently virulence/pathogenicity levels, created a score, based on a qualitative and quantitative evaluation, which ranged from 0 to +3, where 0 is the absence of symptoms/changes, and the other scores are the representation of a gradual increase in symptoms up to +3, which represents the maximum of the alterations found in this experiment. Table 5 shows the total scores for each isolate tested here. 

## 4. Discussion

Many authors have correlated mating type with increased virulence in different fungal species [20,21,22]. On the other hand, Della Terra et al. [13] stated it is unlikely that mating type idiomorphs are directly associated with virulence in *Sporothrix* spp. since the most virulent isolates harbored opposite mating types. These results are in agreement with our findings in the experiments carried out here. Table 6 shows the attempt to correlate the presence of mating type locus (*MAT*) with the virulence score of our *S. schenckii stricto sensu* isolates. The results demonstrate that there was no correlation since isolates with a high virulence score (IOC 1835) showed the same mating as isolates with a moderate score (IOC 2835; IOC 1275; IOC 2547). Similarly, mating type appears to have no influence on the virulence of *Histoplasma capsulatum* and *Cryptococcus neoformans* var. *neoformans* [23,24].

The virulence score of our *S. schenckii stricto sensu* isolates was created as a result of the experimental inoculation performed in a murine model as part of the project to increase knowledge of these isolates. As previously described, these isolates have remained preserved since the late 1920s at CCFF-IOC. Most of them were obtained from human clinical cases of sporotrichosis in Rio de Janeiro, Brazil, identified as *Sporothrix schenckii* long before the sporotrichosis epidemic started in the late 1990s. We studied and re-identified them as *Sporothrix schenkii sensu stricto* by polyphasic identification [16]. However, as these isolates remained for decades in conditions of microaerobiosis and there was a consequent reduction in the growth rate under mineral oil, they could affect hyphal development, conidia formation and the dimorphic process, so they are important in the success of the fungus against a host. Therefore, we decided to find out whether the preservation time would have altered the invasive capacity of these isolates.

The data presented after subcutaneous inoculation in BALB/c mice showed that the invasive capacity of the IOC 1835 isolate was the highest with a score of 11 points, presenting lesions in different organs, splenomegaly and a greater number of cells recovered from the spleen, among others. This isolate was followed by the IOC 2835, IOC 1275, IOC 2547 isolates with a score of nine, eight, seven, respectively, considering them to have moderate virulence. The IOC 2993 and IOC 1912 isolates with a score of six and five were considered to have low virulence. Finally, the IOC 1799 isolate with a score of two showed a marked attenuation of virulence but was able to infect the animals since it was detected by GMS preparations circulating in the lungs of mice (Table 4; Figure 7). All the isolates studied here had their virulence confirmed at the time of initial isolation, as they came from human patients with sporotrichosis. 

These isolates remained under mineral oil for long periods (64 to 34 years) but did not change their morphology, thermo-tolerance and dimorphic capacity as seen previously. An exception is made for the IOC 2993 isolate, which required the use of a sporulation inducer to stimulate conidial production [16]. However, the ability to produce damage in animals varied among them. The maintenance time under mineral oil does not seem to be a determining factor for the loss/attenuation of virulence, since the IOC 2547 isolate, considered to be of moderate virulence, remained under oil for 64 years while the isolate with attenuated virulence (IOC 1799) remained under oil for a similar time, 59 years. The IOC 1835 isolate, the most virulent, remained under oil for 34 years similar to the preservation time of IOC 2993 isolate (37 years), which showed a low virulence equal to the IOC 1912 isolate that remained under oil for 56 years. It is important to highlight the method of preservation under oil, as in addition to the time that the fungus remains under stressful conditions of oxygen and a nutrient deficit, the moment during the growth cycle of the fungus where the drastic process of the reduction in the vital activities for preservation is crucial for the successful maintenance of the morphological and physiological activities of the fungus [25]. Unfortunately, this moment cannot be verified for the isolates studied here, since there are no records about it in the CCFF. 

Lima et al. [26] studied the virulence of *S. schenckii* isolates maintained at CCFF under mineral oil. Among them were the IOC 1835 and IOC 2993 isolates studied here, and their results showed that the IOC 1835 isolate had its virulence attenuated and the IOC 2993 isolate was the most virulent, which was different from what was seen in this study. We can infer that this difference may be related to the route of inoculation used (intravenous by the authors cited above and subcutaneous by us) and also to the non-reactivation of the isolates when starting the experiments. All isolates studied here were reactivated with passage in vivo so that they could be in conditions as close to equality as possible for the results to be reliable. However, similarities were observed between these two studies regarding the spontaneous healing of the lesions in the animals after weeks of observation. Studies on experimental sporotrichosis have already documented spontaneous healing with a remission of the lesions [27,28]. 

Quantitative modifications of cell wall components, the inability to secrete proteases and express some proteins, as well as temperature sensitivity have been factors related to the loss of virulence in many fungi [29,30]. Previous studies on the in vitro storage of different pathogens have suggested that depending on the method, this may result in a loss of virulence [31,32,33,34]. Although the aim of our study was not to identify and quantify virulence factors, the results could suggest that the isolates had altered the process related to the secretion of molecules with a putative role as tissue-invading factors [35]. 

According to Lima and Borba [36], isolates of *Sporothrix schenckii* survive better under mineral oil than other species of dimorphic fungi such as *Blastomyces dermatitidis* and *Histoplasma capsulatum*. Our results prove once again the resistance of *S. schenckii stricto sensu* isolates to preservation time and mineral oil, without knowing the mechanisms that confer this resistance. However, changes were observed, although difficult and laborious to detect. This alerts us to the fact that long periods of preservation under oil can alter the physiological characteristics of fungi, as described by other authors [33,37,38]. 

## 5. Conclusions

The results presented here emphasize the international principles of the ideal preservation of fungi in collections [39], and the use of new tools that provide more information than those produced at the time of entry, and the preservation of isolates, increasing knowledge of these for the scientific community.

The approaches used in this study showed no correlation between the mating type of *S. schenckii stricto sensu* isolates and their virulence, although we believe that more isolates should be studied, since the existing data are still incipient, combined with an investigation of gene expression. In addition, it was possible to infer different degrees of virulence/pathogenicity among the isolates and point out the existence of variants of *S. schenckii stricto sensu* with attenuated virulence that can provide opportunities, with new technologies for studies on the factors that may be related to the pathogenesis of sporotrichosis.

## Figures and Tables

**Figure 1 microorganisms-11-02335-f001:**
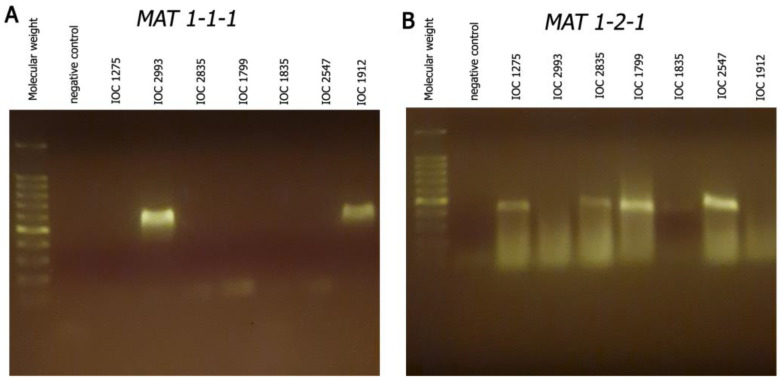
PCR analysis of *Sporothrix schenckii sensu stricto MAT 1-1-1* and *MAT 1-2-1* genes amplified in different isolates. (**A**) Agarose gel showing a ≅ 600 bp fragment in the isolates IOC 2993 and IOC 1912 corresponding to *MAT 1-1-1* gene; (**B**) agarose gel showing a ≅ 450 bp fragment in the isolates IOC 1275, IOC 2835, IOC 1799, IOC 2547 and IOC 1835 corresponding to *MAT 1-2-1* gene. Molecular weight DNA ladder, 100 bp.

**Figure 2 microorganisms-11-02335-f002:**
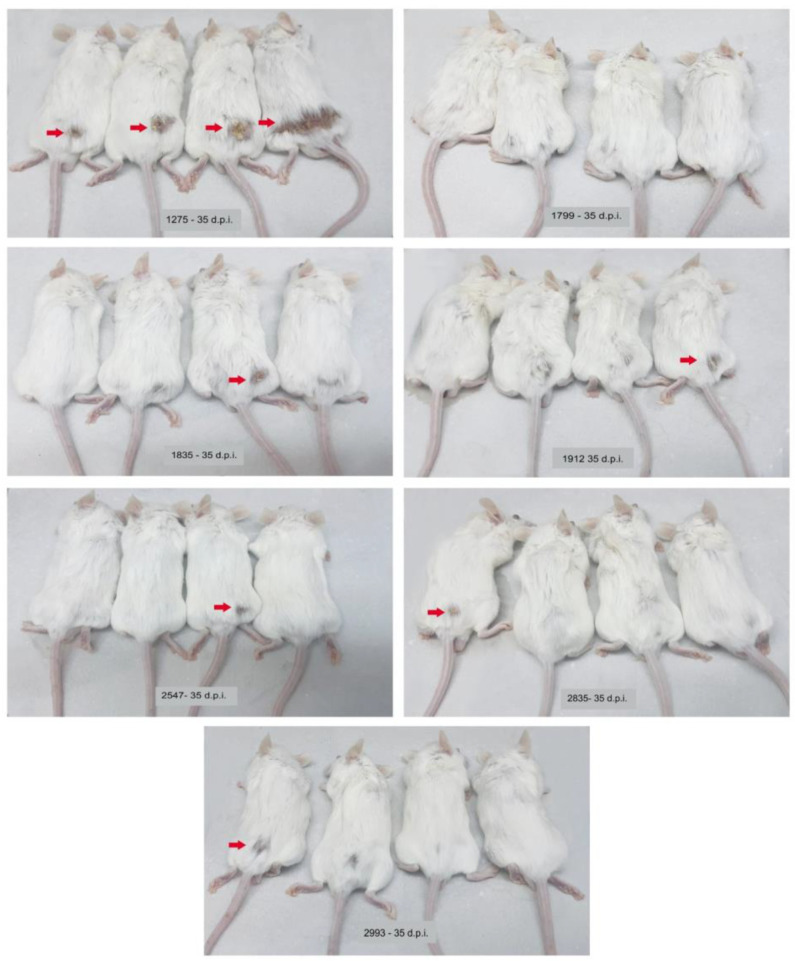
BALB/c mice inoculated with *Sporothrix schenckii stricto sensu* isolates at 35 days of infection. Groups formed by four mice inoculated with the following isolates: IOC 1275, IOC 1799, IOC 1835, IOC 1912, IOC 2547, IOC 2835 and IOC 2993. The red arrows indicate enlarged ulcerated lesions from the point of inoculation.

**Figure 3 microorganisms-11-02335-f003:**
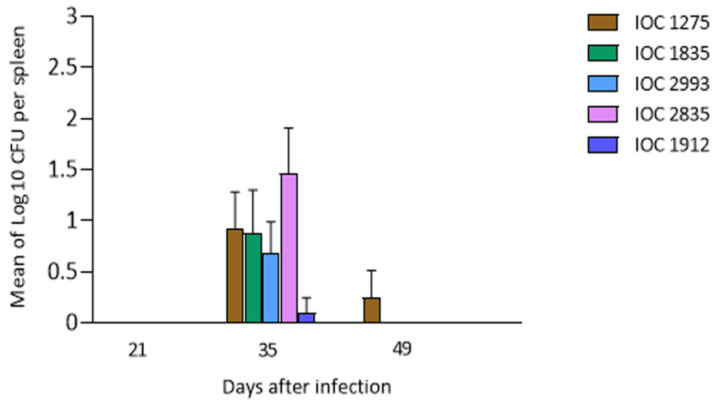
Quantification of viable fungal cells recovered from the spleens of mice infected with *Sporothrix schenckii stricto sensu* isolates (IOC 1275, IOC 1835, IOC 2993, IOC 2835, IOC 1799, IOC 1912, IOC 2547). Each point represents the mean of colony forming units recovered from spleens of four mice euthanized at predetermined times of 21, 35 and 49 days after inoculation.

**Figure 4 microorganisms-11-02335-f004:**
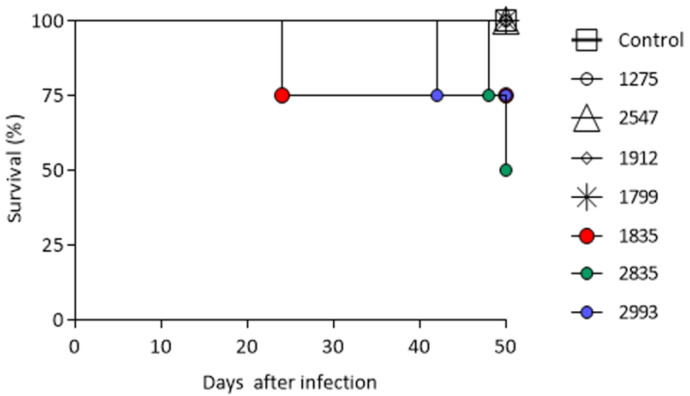
Percentage of survival for mice inoculated with PBS (control group) and mice infected with *Sporothrix schenckii stricto sensu* isolates (IOC 1275, IOC 1835, IOC 2993, IOC 2835, IOC 1799, IOC 1912, IOC 2547) during 50 days of evaluation.

**Figure 5 microorganisms-11-02335-f005:**
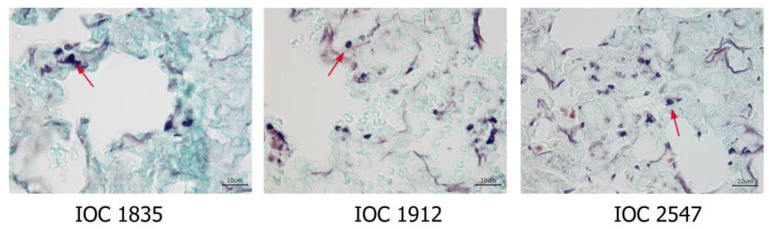
Histological sections of lungs from mice infected with *Sporothrix schenckii stricto sensu* isolates IOC 1835, IOC 1912 and IOC 2547, showing the presence of fungal structures (red arrows) evidenced in Grocott’s methenamine silver (GMS) preparations at 21 days of infection, (1000×).

**Figure 6 microorganisms-11-02335-f006:**
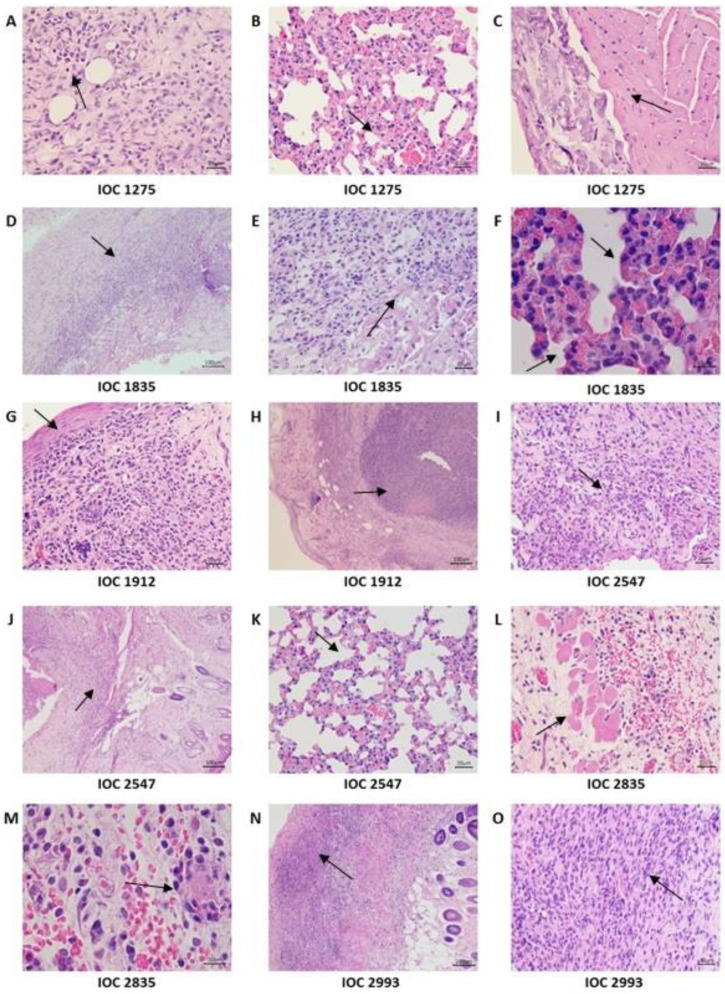
Histological sections of skin, lungs and heart from mice infected with *Sporothrix schenckii stricto sensu* isolates (IOC 1275, IOC 1835, IOC 2547, IOC 1912, IOC 2835, IOC 2993) stained with hematoxylin-eosin at 35 days of infection. Black arrows indicate histological alterations in all organs photographed. IOC 1275 (**A**) skin—moderate suppurative dermatitis with inflammatory infiltrate (400×), (**B**) lung—multifocal and moderate pyogranulomatous pneumonitis with inflammatory infiltrate (400×) and (**C**) heart—focus of dystrophic mineralization (magnification of 400×); IOC 1835 (**D**) skin—marked and diffuse suppurative panniculitis (100×), (**E**) skin—inflammatory infiltrate consisting mainly of neutrophils (400×) and (**F**) lung—moderate and diffuse suppurative pneumonitis (1000×); IOC 1912 (**G**) skin—moderate and diffuse suppurative dermatitis, more intense in the deep dermis (400×) and (**H**) skin—presence of severe and diffuse panniculitis with extensive area of necrosis (100×); IOC 2547 (**I**) skin—inflammatory infiltrate consisting mainly of neutrophils (400×), (**J**) skin—moderate and diffuse dermatitis and panniculitis (100×), and (**K**) lung—discrete and diffuse pyogranulomatous interstitial pneumonitis with areas alveolar wall thickening (400×); IOC 2835 (**L**) skin—subcutaneous tissue showing areas of hemorrhage (400×) and (**M**) skin—discrete and diffuse lymphoplasmacytic inflammatory infiltrate (1000×); IOC 2993 (**N**) skin—focal and discreet suppurative dermatitis (100×) and (**O**) skin—marked and diffuse lymphoplasmacytic panniculitis and inflammatory infiltrate composed mainly of lymphocytes and plasma cells (400×).

**Figure 7 microorganisms-11-02335-f007:**
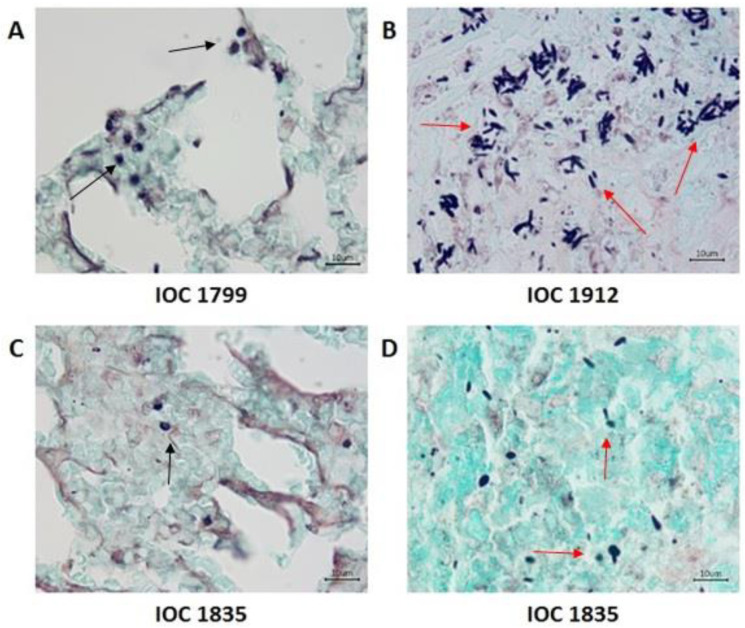
Histological sections of the lung and skin of mice infected with *Sporothrix schenckii stricto sensu* isolates (IOC 1799, IOC 1912, IOC 1835) showing the presence of fungal structures without budding (black arrows) and with budding (red arrows) evidenced in the preparations of Grocott’s methenamine silver (GMS) at 35 days of infection. IOC 1799 (**A**) lung; IOC 1912 (**B**) skin; IOC 1835 (**C**) lung and (**D**) skin (1000×).

**Table 1 microorganisms-11-02335-t001:** *Sporothrix schenckii sensu stricto* isolates preserved in the Culture Collection of Filamentous Fungi of Oswaldo Cruz Institute, Fiocruz (CCFF).

No. of Isolates	Year of Entry into CCFF	Period under Mineral Oil (Years)
IOC 1275	1929	35
IOC 1799	1935	59
IOC 1835	1936	34
IOC 1912	1945	56
IOC 2547	1948	64
IOC 2835	1950	34
IOC 2993	1951	37

**Table 2 microorganisms-11-02335-t002:** Bodyweight of mice inoculated with PBS (control group) and mice infected with IOC 1275, IOC 1835, IOC 2993, IOC 2835, IOC 1799, IOC 1912 and IOC 2547 *Sporothrix schenckii stricto sensu* isolates at 21, 35 and 49 days of infection.

No. of Isolates	Weight (g) ± DP			
	0 Day	21 Day	35 Day	49 Day
PBS	25.00	33.04 ± 0.73	35.04 ± 2.37	34.28 ± 0.91
IOC 1275	25.00	31.19 ± 0.70	33.29 ± 1.43	34.14 ± 0.55
IOC 1835	25.00	33.25 ± 2.19	34.61 ± 0.75	34.88 ± 2.69
IOC 2993	25.00	32.44 ± 3.09	33.07 ± 1.99	33.06 ± 2.12
IOC 2835	25.00	30.98 ± 1.62	31.68 ± 1.91	34.08 ± 0.57
IOC 1799	25.00	32.23 ± 1.16	32.56 ± 1.27	34.75 ± 3.11
IOC 1912	25.00	33.25 ± 2.12	33.69 ± 2.47	33.61 ± 2.51
IOC 2547	25.00	32.14 ± 2.27	31.61 ± 2.10	34.09 ± 2.08

**Table 3 microorganisms-11-02335-t003:** Splenic index of mice infected with *Sporothrix schenckii stricto sensu* isolates after 21, 35 and 49 days of infection, in relation to the inoculated control group with PBS.

Isolates	Days Post Infection		
	21	35	49
IOC 1275	1.01	1.03 ± 0.66	1.02 ± 0.24
IOC 1835	1.71 ± 0.58	1.03 ± 0.23	1.08 ± 0.13
IOC 2993	1.05 ± 0.26	1.05 ± 0.34	1.06 ± 0.13
IOC 2835	1.22 ± 0.43	1.09 ± 0.56	1.10 ± 0.33
IOC 1799	1.07 ± 0.23	1.09 ± 0.21	1.09 ± 0.37
IOC 1912	1.02 ± 0.28	1.09 ± 0.06	1.09 ± 0.12
IOC 2547	1.47 ± 0.46	1.66 ± 0.06	1.23 ± 0.32

Control group = mean value for the relative weight of spleens was equal to one unit.

**Table 4 microorganisms-11-02335-t004:** Histological alterations in the organs of mice inoculated with *Sporothrix schenckii stricto sensu* observed with Hematoxylin-eosin and Grocott’s methenamine silver (GMS) preparations at 21 and 35 days after inoculation.

Isolates	Skin		Heart		Kidney		Lung	
	21	35	21	35	21	35	21	35
IOC 1799	Unseen	Unseen	Unseen	Unseen	Unseen	Unseen	Unseen	+GMS
IOC 1912	Unseen	Moderate and diffuse suppurative dermatitis, more intense in the deep dermis. The inflammatory infiltrate consisted mainly of neutrophils. Presence of marked and diffuse panniculitis with extensive area of necrosis. Extensive destruction of striated skeletal muscle fibers underlying the dermis. +GMS	Unseen	Unseen	Unseen	Unseen	+GMS	Unseen
IOC 1835	Unseen	Marked and diffuse suppurative panniculitis. The infiltrate consisted mainly of neutrophils, with macrophages and lymphocytes in smaller numbers. Areas of necrosis were observed in the subcutaneous tissue. +GMS	Unseen	Unseen	Unseen	Unseen	+GMS	Moderate and diffuse suppurative pneumonitis. Inflammatory infiltrate composed predominantly of neutrophils, with many lymphocytes, with thickening of the alveolar wall. +GMS
IOC 2835	Unseen	Subcutaneous tissue showing areas of hemorrhage and discrete and diffuse lymphoplasmacytic inflammatory infiltrate.	Unseen	Unseen	Unseen	Unseen	Unseen	Unseen
IOC 2993	Unseen	Marked and diffuse lymphoplasmacytic panniculitis. Inflammatory infiltrate composed mainly of lymphocytes and plasma cells. In the other section, focal and discreet suppurative dermatitis was observed, with a predominant infiltrate of neutrophils.	Unseen	Unseen	Unseen	Unseen	Unseen	Unseen
IOC 1275	Unseen	Moderate and diffuse suppurative dermatitis. The inflammatory infiltrate consisted mainly of neutrophils, more intense in the deep dermis. There were areas of spongiosis in the epidermis.	Unseen	Two foci of dystrophic mineralization in the epicardium.	Unseen	Unseen	Unseen	Multifocal and moderate pyogranulomatous pneumonitis. The inflammatory infiltrate was also composed of lymphocytes.
IOC 2547	Unseen	Moderate and diffuse dermatitis and panniculitis with inflammatory infiltrate consisting mainly of neutrophils, more intense in the deep dermis and subcutaneous tissue. Presence of suppurative folliculitis with follicular hyperkeratosis and areas of acanthosis in the epidermis. Additionally, extensive areas of coagulation necrosis were observed in the subcutaneous tissue.	Unseen	Unseen	Unseen	Unseen	+GMS	Discrete and diffuse pyogranulomatous interstitial pneumonitis with areas of alveolar wall thickening. The infiltrate was also made up of lymphocytes and plasma cells.
IOC 2993	Unseen	Marked and diffuse lymphoplasmacytic panniculitis. Inflammatory infiltrate composed mainly of lymphocytes and plasma cells. In the other section, focal and discreet suppurative dermatitis was observed, with a predominant infiltrate of neutrophils.	Unseen	Unseen	Unseen	Unseen	Unseen	Unseen

Unseen = Were not seen tissue alterations and *S. schenckii stricto sensu* yeasts by hematoxylin-eosin and Grocott’s methenamine silver (GMS) preparations. +GMS = Presence of rounded yeast-like structures without/with budding in the parenchyma, suggestive of *S. schenckii stricto sensu.*

**Table 5 microorganisms-11-02335-t005:** Score assigned to *Sporothrix schenckii stricto sensu* isolates based on infection/virulence parameters obtained from the experimental infection of BALB/c mice.

Virulence Parameters									
Isolates	Presence of Macroscopic Lesions	Bodyweight	Clinical Signs	Splenic Index	Fungal Cell Recovery	Survival Rate (%)	Histopathology		Total Score
							Histological Change	GMS+	
IOC 1799	0	0	0	+1	0	0	0	+1	2
IOC 1912	+1	0	+1	+1	+1	0	+1	+1	5
IOC 1835	+2	0	+2	+2	+2	+2	+2	+1	11
IOC 2835	+1	0	+1	+1	+3	+3	+1	0	9
IOC 2993	+1	0	+1	+1	+2	+1	+1	0	6
IOC 1275	+3	0	+3	+1	+2	0	+2	0	8
IOC 2547	+1	0	+1	+3	0	0	+2	+1	7

0 = absence; +1 = minimal changes/low values; +2 = intermediate changes/middle values. +3 = maximum changes/high values.

**Table 6 microorganisms-11-02335-t006:** Correlation between mating-type gene and virulence score of IOC *Sporothrix schenckii stricto sensu* isolates.

Number of Isolates	Mating-Type Gene	Virulence Score
IOC 1275	*MAT 1-2-1*	8
IOC 1799	*MAT 1-2-1*	2
IOC 1835	*MAT 1-2-1*	11
IOC 1912	*MAT 1-1-1*	5
IOC 2547	*MAT 1-2-1*	7
IOC 2835	*MAT 1-2-1*	9
IOC 2993	*MAT 1-1-1*	6

## Data Availability

The data that support the findings of this study are available from the corresponding author upon reasonable request.

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
