# Peer review of "The Mating Type (MAT) and Virulence of Sporothrix schenckii sensu stricto Isolates Maintained in Culture Collection"

_microorganisms, 2023, doi:10.3390/microorganisms11092335_

Round 1

Reviewer 1 Report

Check the italics in fungal names

May could add to abstract: new technologies for studies of factors can improve knowldege of pathogenesis

Author Response

Check the italics in fungal names

May could add to abstract: new technologies for studies of factors can improve knowldege of pathogenesis

Authors: it was added to te abstract (lines 23-25).

Reviewer 2 Report

- The title doesn't really reflect the work. No correlation analysis is performed and MAT1-1-1 only contains 2 isolates which is not enough for this work. The word "attempt" also is not a great start for this piece of work. Lastly, aren't all strains maintained in a collection of some sort, even if it is someones personal collection? Doesn't add much to the title.

- L29-32 The One fungus, One name speciation nomenclature should be used here, stages are not specifically named any more.

- L34 Hyperendemicity has been described in more regions than Brazil. Think China, Australia, Peru. Shortsighted to name just one country. 

- L35: "the MAT locus" by MAT loci. Some fungi have multiple.

- L46: invasive aspergillosis (typo), and this work has been superseded by Rhodes et al 2022 "Population genomics" shows there is no difference.

- L73: extraction of DNA is not by chloroform:isoamyl. This is to purify and clean up DNA, what method is used to break open cells? Bead beating, grinding, what buffer? More important.

- L131-133: please describe timing of stainings and where purchased from? Was this a kit and most important for GMS it matters what initial steps were performed for staining.

- L138: If Kruskal-Wallis non-parametric is used, then reports and graphs should all show median data as mean is not an useful metric for non-parametric data.

- Materials and methods: A section on ethics needs to be included. Where submitted and who approved these methodologies? 

- Table 2: it is impossible that all animals weighed exactly 25g on day 0. What was median weight per group and the deviation?

- Figure 3: Labels should be in English. And this colour palette is not colour blind friendly. Can't distinguish the different bars.

- L213: italic of S. schenckii.

- Figure 4: Can't distinguish between isolates and what line is what. Easier to separate some plots.

- Table 4: Can be in supplementary, not very useful to have such a large table with not an awful lot of information.

-Figure 5: these images are too low in quality to be able to say something useful about it.

- Table 5: are these scores subjective or generally used? A ref is needed if so. This seems rather subjective..

- Minor spell check and grammar check required.

Author Response

- The title doesn't really reflect the work. No correlation analysis is performed and MAT1-1-1 only contains 2 isolates which is not enough for this work. The word "attempt" also is not a great start for this piece of work. Lastly, aren't all strains maintained in a collection of some sort, even if it is someones personal collection? Doesn't add much to the title.

Authors: We changed the title as you and the other reviewer suggested. “The mating type (MAT) and virulence of Sporothrix schenckii sensu stricto isolates maintained in culture collection.

- L29-32 The One fungus, One name speciation nomenclature should be used here, stages are not specifically named any more.

Authors: It was corrected.

- L34 Hyperendemicity has been described in more regions than Brazil. Think China, Australia, Peru. Shortsighted to name just one country. 

Authors: We agree with your comment, however, we cited Brazil because all the isolates, except for IOC 1799, were originally isolated in Rio de Janeiro/ Brazil. In this state, sporotrichosis has compulsory notification, as a result of the large number of human and feline cases due to the zoonotic transmission profile of the disease. Nevertheless, we cited “ and the world” in this corrected version.

- L35: "the MAT locus" by MAT loci. Some fungi have multiple.

Authors: It was corrected.

- L46: invasive aspergillosis (typo), and this work has been superseded by Rhodes et al 2022 "Population genomics" shows there is no difference.

Authors: Thanks for your update. We removed the sentence so as not to mislead the reader.

- L73: extraction of DNA is not by chloroform:isoamyl. This is to purify and clean up DNA, what method is used to break open cells? Bead beating, grinding, what buffer? More important.

Authors: The method used to extract DNA was  grinding in liquid nitrogen, with TES buffer. It was added to the methodology (lines 83-86)

- L131-133: please describe timing of stainings and where purchased from? Was this a kit and most important for GMS it matters what initial steps were performed for staining.

Authors: It was not used stainig kits. The protocols and company where the stains were purchased, are described in lines 148-164.

- L138: If Kruskal-Wallis non-parametric is used, then reports and graphs should all show median data as mean is not an useful metric for non-parametric data.

Authors: Thank you for this commentary. It was correct in the text, since was used ANOVA to compare the means of the eight independent groups in three points of observation.

- Materials and methods: A section on ethics needs to be included. Where submitted and who approved these methodologies? 

Authors: It was included in the item 2.3.4, lines166-169

- Table 2: it is impossible that all animals weighed exactly 25g on day 0. What was median weight per group and the deviation?

Authors: We have assumed an average of 25 g because when we order mice from the breeding center, we specify age and weight. In this way, these animals are brought to us according to these parameters.

- Figure 3: Labels should be in English. And this colour palette is not colour blind friendly. Can't distinguish the different bars.

Authors: The figure was adequate to your request

- L213: italic of S. schenckii.

- Figure 4: Can't distinguish between isolates and what line is what. Easier to separate some plots.

Authors: The figure was adequate to your request

- Table 4: Can be in supplementary, not very useful to have such a large table with not an awful lot of information.

Authors: Thanks for your comment and allow us to disagree with you. We believe that the way the results were presented, in tabular form, provides more clarity when comparing results between isolates, facilitating the reader’s understanding and should be in the main body of the text.

-Figure 5: these images are too low in quality to be able to say something useful about it.

Authors: The figure was adequate to your request

- Table 5: are these scores subjective or generally used? A ref is needed if so. This seems rather subjective.

Authors: Experimental animal studies use scores as a guide for several parameters, including humane endpoints and disease severity. In virulence studies, some criteria are used to compare the pathogenic potential between strains or species of microorganisms.

These criteria have been used by our group for decades, and, in order to present them in a more didactic way to the reader, we established these scores to infer more or less aggressiveness among the isolates.

Similar studies have already been conducted, a long time ago, such as the study made by Mitchson et al (https://doi.org/10.1016/s0041-3879(60)80019-0), in which the virulence of tuberculosis bacilli infecting guinea pigs was estimated. In this study, mortality, histological changes were observed and colony-forming units were quantified, with virulence scores being attributed to each of these criteria.

Reviewer 3 Report

Thank you for a great study. I have several questions.

1. Mating type of stain IOC 1912 and IOC 1835

line 141-153, IOC 1912 had MAT 1-1-1 and MAT 1-2-1, IOC 1835 did not have MAT 1-1-1, MAT 1-2-1. In table 6, IOC 1912 was MAT 1-1-1, IOC 1835 was MAT 1-2-1. What is true?

2.Mating type and virulence

Virulence score of 4 stains of MAT 1-2-1 were higher than 2 strains of MAT 1-1-1 except IOC 1799. Do you think it is not significant difference?

3. Title

The number of strains was too small to find the correlation between mating type and virulence. Your main discussion was preservation time and virulence, So I recommend changing your title as below.  

Molecular and animal experimentation approaches in the correlation attempt between The mating type (MAT) and virulence of Sporothrix schenckii sensu stricto isolates maintained in culture collection.

Author Response

Thank you for a great study. I have several questions.

  1. Mating type of stain IOC 1912 and IOC 1835

line 141-153, IOC 1912 had MAT 1-1-1 and MAT 1-2-1, IOC 1835 did not have MAT 1-1-1, MAT 1-2-1. In table 6, IOC 1912 was MAT 1-1-1, IOC 1835 was MAT 1-2-1. What is true?

Authors: We apologize for this this mistake. Actually IOC 1912 had MAT 1-1-1 and IOC 1835 was MAT 1-2-1. This has already been corrected in the text (lines 185 and 190).

 2.Mating type and virulence

Virulence score of 4 stains of MAT 1-2-1 were higher than 2 strains of MAT 1-1-1 except IOC 1799. Do you think it is not significant difference?

Authors: We think it is not significant. In fact the virulence score of the 4 strains of MAT 1-2-1 ranged from 11 to 7 and was higher than the 2 strains of MAT 1-1-1 which ranged from 5 to 6. However, we did not consider it to be significant because the diferences of a strain with a virulence score of 6 are litlle compared to a strain with a virulence score of 7. They are very close to each other. We would consider, in principle, thinking of significance it the strain with the highest virulence score (11) presented MAT different from the others, in addition, of course, to highlight the need for future confirmation due to the small number of samples used in this study.

  1. Title

The number of strains was too small to find the correlation between mating type and virulence. Your main discussion was preservation time and virulence, So I recommend changing your title as below.  

 Molecular and animal experimentation approaches in the correlation attempt between The mating type (MAT) and virulence of Sporothrix schenckii sensu stricto isolates maintained in culture collection.

Authors: We changed the title as you suggested. “The mating type (MAT) and virulence of Sporothrix schenckii sensu stricto isolates maintained in culture collection.

Round 2

Reviewer 2 Report

All my queries and questions have been answered. I'm happy with the response.